# Auto-Panoptic: Cooperative Multi-Component Architecture Search for Panoptic Segmentation

**Yangxin Wu**[1], **Gengwei Zhang**[1], **Hang Xu**[2], **Xiaodan Liang**[1,3], and **Liang Lin**[1,3*]

[1]*Sun Yat-sen University*, [2]*Huawei Noah's Ark Lab*, [3]*DarkMatter AI Research*
wuyx29@mail2.sysu.edu.cn, {zgwdavid, chromexbjxh, xdliang328}@gmail.com, linliang@ieee.org

## Abstract

Panoptic segmentation is posed as a new popular test-bed for the state-of-the-art holistic scene understanding methods with the requirement of simultaneously segmenting both foreground *things* and background *stuff*. The state-of-the-art panoptic segmentation network exhibits high structural complexity in different network components, i.e. backbone, proposal-based foreground branch, segmentation-based background branch, and feature fusion module across branches, which heavily relies on expert knowledge and tedious trials. In this work, we propose an efficient, cooperative and highly automated framework to simultaneously search for all main components including backbone, segmentation branches, and feature fusion module in a unified panoptic segmentation pipeline based on the prevailing one-shot Network Architecture Search (NAS) paradigm. Notably, we extend the common single-task NAS into the multi-component scenario by taking the advantage of the newly proposed intra-modular search space and problem-oriented inter-modular search space, which helps us to obtain an optimal network architecture that not only performs well in both instance segmentation and semantic segmentation tasks but also be aware of the reciprocal relations between foreground *things* and background *stuff* classes. To relieve the vast computation burden incurred by applying NAS to complicated network architectures, we present a novel path-priority greedy search policy to find a robust, transferrable architecture with significantly reduced searching overhead. Our searched architecture, namely Auto-Panoptic, achieves the new state-of-the-art on the challenging COCO and ADE20K benchmarks. Moreover, extensive experiments are conducted to demonstrate the effectiveness of path-priority policy and transferability of Auto-Panoptic across different datasets. Codes and models are available at: https://github.com/Jacobew/AutoPanoptic.

## 1 Introduction

Recently the community has witnessed great progress in semantic segmentation and instance segmentation. However, a more desirable system is expected to directly perform image holistic understanding about all instances and background segments in one feedforward step. To bridge the chasm between the understanding of foreground *things* and background *stuff*, panoptic segmentation [21] becomes a new test-bed on which the state-of-the-art methods can validate their capability in simultaneously parsing the foreground objects at the instance level and the background contents at the semantic level.

Some attempts [20, 22] try to tackle two tasks via stitching the proposal-based foreground branch and segmentation-based background branch without compromising the accuracy of instance and semantic

---

segmentation. While conceptually straightforward, they overlook the underlying relations between foreground and background, and is thus subject to limited performance gain. Some later approaches [40, 28, 23, 22, 38, 43] try to explicitly model this relationship in a learnable fashion. UPSNet [40] utilizes a panoptic head to jointly optimize the outputs of instance and semantic segmentation. BGRNet [38] proposes bidirectional feature fusion at the proposal and class level based on a graph neural network. However, these approaches heavily rely on carefully hand-tuned architectures and require complicated processes such as RoI-Flatten [22], RoI-Upsample [23], mask pruning process [40], and so on. Moreover, the designs of key components, e.g., backbone and heads, and different combinations of them can have a significant impact on the performance of a panoptic segmentation model, which is also observed in other fine-grained vision tasks like object detection [44, 14, 35]. This manifests the necessity of developing a well-designed architecture for panoptic segmentation that has optimal combinations of different components with minimal human efforts.

In this paper, inspired by the advances achieved by Network Architecture Search (NAS) on fundamental vision tasks [27, 2, 36, 41], we aim to explore an efficient foreground-background cooperative architecture automatically tailored for panoptic segmentation, namely Auto-Panoptic. Unlike previous work that only exploits a specific part of the network, our Auto-Panoptic extends the common NAS paradigm into the multi-component scenario and manages to search for all key components including backbone, segmentation branches and feature fusion module in a unified pipeline.

First, to get an optimal combination of different key components in the panoptic segmentation pipeline, Auto-Panoptic learns customized intra-modular structures of the backbone, instance segmentation branch and semantic segmentation branch in an end-to-end fashion. Second, to fully utilize the complementary information provided by foreground objects and background stuff, the inter-modular search space is proposed to establish the correlations between two separate branches. Finally, regarding the severe convergence inefficiency in the architecture search phase, we disentangle different choice paths in supernet and propose a novel path-priority search policy to perform path-level evaluation, which significantly reduces searching overhead without compromising the accuracy.

In summary, our contributions are summarized as follows:

- To the best of our knowledge, we make the first effort to search for the whole structure for panoptic segmentation, which helps us to derive a robust, transferable, high performing network architecture with minimal human efforts.
- We reformulate the search space of panoptic segmentation networks as a combination of an efficient intra-modular search space and a problem-oriented inter-modular search space, which helps Auto-Panoptic not only perform well in two separate tasks but also be aware of the reciprocal relations between foreground *things* and background *stuff*.
- Being confronted with the vast search space in the architecture search phase, we propose a novel greedy approach, named the Path-Priority Search Policy, that can significantly reduce searching overhead in searching for a high performing architecture.
- Our Auto-Panoptic achieves new state-of-the-art results on two challenging benchmarks, i.e., COCO and ADE20K and we conduct extensive experiments to demonstrate the robustness of the searched architecture and the effectiveness of our framework.

## 2  Related Work

**Panoptic Segmentation.** Recent studies on panoptic segmentation mainly fall into three categories: combining methods from separate models [21], simple multi-branch schemes [11, 42], and inter-modular fusion methods [23, 40, 38]. Panoptic FPN [20] combines Mask R-CNN [16] with a semantic segmentation branch to build multi-scale features and predict fine-grained panoptic output. UPSNet [40] uses a panoptic head to tackle the inconsistency between instance segmentation and semantic segmentation. BGRNet [38] proposes bidirectional feature fusion to jointly optimize the results from separate branches. In general, the architecture design and the complementarity of foreground and background features are crucial for improving panoptic segmentation quality.

**Neural Architecture Search.** Neural architecture search (NAS) aims at automating the labor-intensive architecture engineering process and has shown promising results on fundamental computer vision tasks like image classification [1, 27, 39], object detection [32, 36, 13] and semantic segmentation [26, 31]. Many weight-sharing approaches [4, 37, 27] optimize the architecture parameters and supernet weights simultaneously based on a continuous relaxation of the architecture search space.

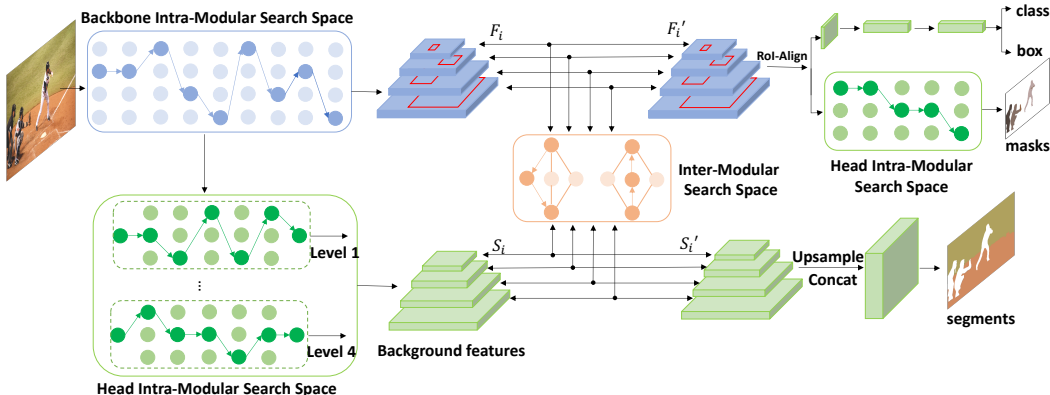

Figure 1: The network architecture of Auto-Panoptic. We search for backbone, mask head in instance segmentation branch, subnet in semantic segmentation branch, and inter-modular fusion module.

This approach leads to a huge memory footprint due to the non-negligible architecture parameters [12]. Moreover, the shared weights across a wide range of architectures become ineffective and unpredictable in the ranking process since the architecture parameters and supernet weights are deeply coupled and performance collapse is often observed during optimization [9, 15, 8]. As a new paradigm, single-path one-shot NAS no longer relies on continuous relaxation in supernet training and the training process is decoupled from architecture search [2, 15, 8, 32]. [15] uses a uniform sampling method to train a weight-sharing supernet where each single path architecture is uniformly sampled and get trained equally. FairNAS [8] takes it a step further and enforces the constraint of strict fairness to eliminate the unfairness between frequently trained models and poorly trained ones.

# 3 Methodology

## 3.1 Motivation and Overview

Typically, our network is expected to not only perform well in both instance and semantic segmentation, but also be aware of the reciprocal relations between *things* and *stuff* for further performance boost. This poses challenges to the design of a panoptic segmentation pipeline in two aspects. First, since the overall architecture can greatly influence the performance, the key components of network should be compatible with each other. Due to the high structural complexity of conventional pipelines, it heavily relies on the expert knowledge and tedious trials to derive an optimal configuration by hand. Second, to fully leverage the complementary cues between foreground objects and background stuff, a cross-modular feature fusion module is desired to bridge the chasm between separate branches.

Despite the generality and efficiency of single-path NAS approaches, it is nontrivial to apply them to the panoptic pipeline due to its multi-component nature and the severe convergence inefficiency incurred by vast search space in the architecture search phase. Unlike previous approaches that only exploit one specific part of a network, we make the first attempt to simultaneously search for all key components including the backbone, instance segmentation branch, semantic segmentation branch, and cross-modular feature fusion module in an end-to-end fashion.

For searching intra-modular structures, we use efficient search spaces according to the preferences of different components in network to fulfill their potential. For searching cross-modular feature fusion module, we propose a novel problem-oriented inter-modular search space to mutually calibrate the features in each branch bidirectionally. Crucially, being confronted with the vast search space presented in the multi-component scenario, we propose a greedy approach named Path-Priority Search Policy to explore competitive models in supernet with significantly reduced searching overhead compared to the evolutionary algorithm. The network architecture of Auto-Panoptic is shown in Figure 1.

## 3.2 Multi-Component Architecture Search

### 3.2.1 Searching for Intra-Modular Structures

Empirically, different components prefer different operators due to their functional distinctions [14, 44]. For example, operators with skip connection and bottleneck structure is preferred in backbone to ease the gradient propagation in deep network. While operators in head are expected to enlarge the receptive field to boost the performance in fine-grained vision tasks. To this end, we customized search spaces for the backbone and head to improve the overall searching efficiency.

**Backbone.** We search for an efficient backbone based on the ShuffleNetv2 block [30], a kind of lightweight convolution architectures that involve channel shuffle operations. For a fair comparison, we keep the parameters and flops of searched architectures the same as that of ResNet [17], which is a commonly used backbone in many fine-grained vision tasks.

**Instance Segmentation Branch.** Given the predicted proposal from the box head, the mask head outputs a binary mask based on the feature after pooling. The architecture of the mask head is crucial to predict high-quality masks and thus we search for a sequence of diverse convolution layers beyond its plain architecture in Mask R-CNN[16] to improve the segmentation quality of foreground instances.

**Semantic Segmentation Branch.** Multi-level features are instrumental for many fine-grained tasks, especially panoptic segmentation, where features from different levels contain diverse information of different granularity. Most previous works [23, 40, 38] on panoptic segmentation share a similar architecture for semantic segmentation, i.e., the multi-level features from backbone are first refined by a series of subnets at different levels and are then merged to produce final semantic segmentation outputs. Thus, we retain this basic structure and search for the architectures of subnets. In particular, we search for a customized three-layer subnet that gradually refines features at each level.

**Search Space Design.** For backbone search, we use the same search space as DetNAS [32] to search beyond Shufflenetv2 block. For head search, we include depth-wise separable convolution [18], atrous convolution [5], and deformable convolution [46] to enlarge the receptive field and maintain modest number of parameters. The adopted search space is listed in Table 1 and more details can be found in Supplementary Materials.

### 3.2.2 Searching for Inter-Modular Attention

Previous methods [28, 40, 23, 22] use complicated fusion modules to remap the outputs from two branches to a canvas which is of the same size as the semantic segmentation feature map to align foreground masks with background stuff, and thus introduces a considerable amount of parameters and high memory occupation. In contrast, Auto-Panoptic aims to capture the reciprocal relations between *things* and *stuff* at the feature level rather than the mask level, which allows us to circumvent the complicated process of instances with variable number and size.

Intuitively, the multi-level features in the Region Proposal Network (RPN) [34] and semantic segmentation branch contain rich foreground and background information, respectively. Based on this observation, we introduce a bidirectional attention mechanism between features of different levels to fully leverage this kind of complementarity at a fine granularity. Inspired by SENet [19], we find it beneficial to capture channel-wise dependencies across branches and adaptively recalibrate their feature responses bidirectionally. As is shown in Figure 1, we establish an inter-modular search space between the features of different levels from the RPN and semantic segmentation branch. Formally, given the multi-level features $\{F_i\}_{i=1:5}$ from the RPN and $\{S_i\}_{i=1:4}$ from the semantic segmentation branch, the attention vector from foreground to background at the $i$-th level can be obtained by:

$$a_{f \to b}^i = f_{ex}(f_{sq}(F)) = \delta_1(W_1 \delta_2(W_2 z_i)), \tag{1}$$

where $F$ is the concatenation of $\{F_i\}_{i=1:5}$ after bilinear upsample the coarser feature maps, $\delta_1$ and $\delta_2$ are non-linear activation functions, $W_1 \in \mathbb{R}^{C_s \times C_s/r}$, $W_2 \in \mathbb{R}^{C_s/r \times C_r}$, $z_i \in \mathbb{R}^{C_r}$ is the channel-wise statistics generated by applying global average pooling to $F$, $r$ is the parameter of bottleneck between two fully connected layers. The attention vectors from background to foreground features, i.e., $\{a_{b \to f}^i\}_{i=1:4}$, can be obtained via similar operations. The attention vectors are used to dynamically calibrate the features at each level $i$ in both branches via residual connection:

$$F_i' = F_i + a_{b \to f}^i \otimes F_i, \quad S_i' = S_i + a_{f \to b}^i \otimes S_i, \tag{2}$$

| Component | Backbone | Head | |
|---|---|---|---|
| Intra-Modular | Shuffle3x3 Shuffle 5x5 Shuffle7x7 Xception3x3 | DWConv3x3  DWConv5x5 ATConv3x3  ATConv5x5 DFConv3x3  DFConv5x5 | |
| Inter-Modular | reduction ratio=(4,8,16) | | |

Table 1: Search spaces adopted in Auto-Panoptic. DWConv indicates depth-wise separable convolution. ATConv indicates atrous convolution. DFConv indicates deformable convolution.

where $\otimes$ denotes channel-wise multiplication, $+$ denotes element-wise addition. With this fine-grained level-wise attention mechanism, we enable input-specific and semantic-aware feature calibration across branches and incurs only minor increase in parameters due to the highly efficient bottleneck structure. We search for $r$, the reduction ratio of bottleneck, in inter-modular search space to explicitly control the model complexity and generalization ability.

## 3.3 Auto-Panoptic Search Algorithm

As described in Section 2, the widely disseminated DARTS [27] is observed to encounter performance collapse [9] during optimization and is not memory-friendly due to the non-negligible architecture parameters. Therefore, we resort to single-path one-shot NAS to eliminate architecture parameters in the training process. Formally, the search space is encoded in a weight-sharing supernet $\mathcal{N}(\mathcal{A}, W)$ that encompasses all candidate networks, where $\mathcal{A}$ denotes the architecture search space and $W$ denotes the weights of the supernet. A model $a \in \mathcal{A}$ is sampled from the supernet in each forward step and its weights get optimized in back-propagation. Single-path one-shot NAS aims to find the optimal architecture that minimizes the validation loss via executing some architecture search policy (described in Section 3.4) on the supernet with optimized weights $W_{\mathcal{A}}^*$:

$$\min_{a \in \mathcal{A}} \mathcal{L}_{val}\left(\mathcal{N}\left(a, W_{\mathcal{A}}^*(a)\right)\right) \text{ s.t. } W_{\mathcal{A}}^* = \arg\min_{W} \mathcal{L}_{\text{train}}(\mathcal{N}(\mathcal{A}, W), \mathcal{S}), \tag{3}$$

where $\mathcal{L}_{\text{train}}(\cdot)$, $\mathcal{L}_{\text{val}}(\cdot)$ denote the loss function on the training and validation set, and $\mathcal{S}$ denotes the sampling strategy. We adopt a fair sampling strategy that complies strictly with *Strict Fairness* [8], i.e., we implement $\mathcal{S}$ as a uniform sampling strategy without replacement. Under this premise, the parameters of every choice block in both intra-modular and inter-modular search spaces are optimized the same number of times after the training process. Our work is orthogonal to existing NAS methods, and advances in the literature of single-path NAS may also be applicable in our framework. The details of the Auto-Panoptic supernet training strategy are depicted in Algorithm 1. The function PERMUTE$(\cdot, \cdot)$ returns a sampling of choice paths that meets the criteria that each choice path gets activated and updated without repetition in each training step. It is noteworthy that network parameters are updated once after several times of back-propagation, which eliminates the bias caused by different training orders of choice paths.

## 3.4 Path-Priority Search Policy

Once the supernet has converged, architecture search is performed to pick out a high-performing model. Some previous works [3, 2] use random search to obtain favorable models, which is not effective since the search space can be too large to explore in a stochastic pattern. Other works [8, 15] adopt powerful Evolutionary Algorithm (EA) to discover promising models via individual evolution.

**Limitations of EA.** The powerful exploration ability of EA in a vast search space comes at the cost of maintaining a large population that evolves over time. Hyper-parameters, e.g., *population size*, *max iterations*, *crossover probability*, *mutation probability*, need careful tuning to achieve good performance. This results in substantial overhead in fine-grained tasks like panoptic segmentation, which is overlooked or not discussed by other works. Consider a population of size 50 that evolves over 20 generations, which is a common setting in many one-shot [32, 15] approaches. Typically, it takes roughly 8 minutes to finish the evaluation on COCO [25] val set on 8 Tesla V100 GPUs. Accordingly, EA needs 5.6 days to evaluate 1000 models to ensure the diversity and convergence of the population, which is prohibitively expensive.

**Path-Priority Search.** The underlying reason for this inefficiency is that the entire model rather than the choice path is rated in evaluation. As can be seen, the number of possible models increases exponentially with the number of search layers, while the size of the single-path model grows linearly.

The overall search space in Auto-Panoptic includes more than $6.7 \times 10^{33}$ models, which is either inefficient or expensive for random search and EA to explore. In our framework, since all choice paths are trained equally due to the fair training strategy, the supernet is robust to the co-adaptation of components. Thus it is possible to rate different choice paths rather than an entire model in order to reduce the complexity of architecture search. Based on the above observations, we propose Path-Priority Search Policy with linear complexity for efficient architecture search.

The details of Path-Priority Search Policy are depicted in Algorithm 1. Note that for each testing cycle, each choice path has exactly the same number of occurrences. When rating a choice path, we choose Panoptic Quality (PQ) of the model that this path appears as the fitness of it based on a greedy assumption: when a model is good, then its paths are good as well. We assign scores of all choice paths in $model_i$ based on its fitness by a linear function: $score_i = K - rank_i$, where $K$ is a constant, and accumulate the scores over $T$ cycles. Given the leaderboard recording the scores of all choice paths, we simply choose the path with the highest score to build up the best model.

Being conceptually simple, our Path-Priority Search Policy has several prominent advantages. First, it suffices to find a high performing model with much fewer enumerated models. Second, it is easy to get parallel acceleration since the evaluation of each testing cycle is independent. Last but not least, it introduces no hyper-parameters other than the number of evaluating cycles $T$. We further demonstrate the effectiveness of this approach in Section 4.4.

---

**Algorithm 1** Auto-Panoptic Supernet Training Strategy and details of Path-Priority Search Policy.

**Input:** Training iterations $N$, dataloader $D$, intra-modular search layer depth in backbone/heads $L_1/L_1'$, inter-modular search layer depth $L_2$, choice paths per intra-modular search layer in backbone/heads $P_1/P_1'$, choice paths per inter-modular search layer $P_2$, evaluating cycles $T$, number of enumerated models per cycle $E$, leaderboard of all choice paths $L$.

**Output:** Best model found.

    **for** $i = 1$ to $N$ **do**                            # Supernet Training Process.
        **for** *image, labels* **in** $D$ **do**
            $sample_i$ = PERMUTE($\{P_1, P_1', P_2\},\{L_1, L_1', L_2\}$);
            Build $model_i$ from $sample_i$ and accumulate gradients w.r.t $Loss(model_i(image), labels)$;
            **if** $i$ is divisible by $|P_i|_{i=1:3}$ **then**
                Update parameters by accumulated gradients;
                Zero gradients;
    **for** $i = 1$ to $T$ **do**                            # Architecture Search Phase.
        **for** $j = 1$ to $E$ **do**
            $sample_j^i$ = PERMUTE($\{P_1, P_1', P_2\},\{L_1, L_1', L_2\}$);
            Build $model_j^i$ from $sample_j^i$;
            $fitness_j$ = EVALUATE($model_j^i$);
        $\{score_j, rank_j\}_{j=1:E}$ = RANK($\{fitness_j\}_{j=1:E}$)
        UPDATE($L,\{score_j\}_{j=1:E}$)
    $best\_model$ = SELECT($L$)
    **return** $best\_model$

---

## 4 Experimental Results

### 4.1 Implementation Details

We conduct all experiments on 8 Tesla V100 GPUs using PyTorch [33]. We adopt a shared Feature Pyramid Network [24] to produce multi-level features. We randomly select 5k images and 2k images from the training set of COCO and ADE20K as our validation set in the architecture search phase. Following [32], we first pretrain the supernet backbone on ImageNet using a batch size of 1024 for 300k iterations. Before architecture search, the supernet is trained for 12 epochs and 24 epochs on COCO and ADE20K, respectively, using mini-batch SGD with a weight decay of 0.0001 and a momentum of 0.9. The initial learning rate is 0.02 and is divided by 10 at the $8^{th}$ and $11^{th}$ epoch for COCO, $16^{th}$ and $22^{th}$ epoch for ADE20K, respectively. After architecture search, we retrain the best model under the same training schedule as supernet. In our settings, $L_1 = 40, L_1' = 7, L_2 = 9$, and

the overall search space includes over $4^{40} \times 6^7 \times 3^9 \approx 6.7 \times 10^{33}$ candidate architectures. $\delta_1$ and $\delta_2$ are ReLU and 1-Sigmoid respectively. For architecture search, we find $T = 5, E = 12$ suffices to find a satisfactory model, which makes the total number of enumerated models in the Path-Priority Search Policy roughly equals to that of one generation in EA (60 vs 50). We do not freeze any layer in backbone when perform super pretraining on ImageNet and supernet finetuning on COCO/ADE20K. More details can be found in Supplementary Materials.

## 4.2 Datasets and Evaluation Metrics

We conduct experiments on **MS-COCO** [25] and **ADE20K** [45]. MS-COCO is one of the most challenging datasets consisting of 115k images for the training set, 5k images for the validation set, and 20k images for the test-dev, and there are 80 *things* and 53 *stuff* classes in total. ADE20K is a dataset with more than 20k scene-centric images annotated with objects and object parts. It consists of 20k images for training and 2k images for validation, with 100 *things* and 50 *stuff* classes. We adopt the standard evaluation metric, i.e., Panoptic Quality(PQ), for panoptic segmentation [21]. PQ can be viewed as the multiplication of two quality metrics, segmentation quality(SQ) and recognition quality(RQ).

Table 2: Comparison on COCO val set and ADE val set. DF Conv. indicates the use of deformable convolution [10]. Panoptic-FPN-D is the deformable counterpart of Panoptic-FPN. UPSNet-C denotes UPSNet without the panoptic head. † means our implementation. - means inapplicable.

| | Method | DF Conv. | PQ | $PQ^{Th}$ | $PQ^{St}$ | SQ | RQ |
|---|---|---|---|---|---|---|---|
| MS-COCO | Panoptic-FPN [20] | | 39.0 | 45.9 | 28.7 | - | - |
| | Panoptic-FPN-D† | ✓ | 39.9 | 46.9 | 29.3 | 78.4 | 49.0 |
| | AUNet [23] | | 39.6 | 49.1 | 25.2 | - | - |
| | OANet [29] | | 39.0 | 48.3 | 26.6 | 77.1 | 47.8 |
| | UPSNet-C [40] | ✓ | 41.5 | 47.5 | 32.6 | 79.1 | 50.9 |
| | UPSNet [40] | ✓ | 42.5 | 48.5 | 33.4 | 78.0 | 52.5 |
| | SpatialFlow [6] | ✓ | 40.9 | 46.8 | 31.9 | - | - |
| | BANet [7] | ✓ | 43.0 | 50.5 | 31.8 | 79.0 | 52.8 |
| | BGRNet [38] | ✓ | 43.2 | 49.8 | 33.4 | **79.1** | 52.7 |
| | SOGNet [43] | ✓ | 43.7 | 50.6 | 33.2 | 78.7 | 53.5 |
| | Auto-Panoptic | ✓ | **44.8** | **51.4** | **35.0** | 78.9 | **54.5** |
| ADE20K | Panoptic-FPN† [20] | | 29.3 | 32.5 | 22.9 | 69.5 | 36.9 |
| | Panoptic-FPN-D† | ✓ | 30.1 | 33.1 | 24.0 | 69.3 | 37.5 |
| | BGRNet [38] | ✓ | 31.8 | **34.1** | 27.3 | 72.2 | 39.5 |
| | Auto-Panoptic | ✓ | **32.4** | 33.5 | **30.2** | **74.4** | **40.3** |

## 4.3 Comparisons with State-of-the-Art

Comparisons with recent state-of-the-art methods on COCO and ADE20K are listed in Table 2. To eliminate the impact of deformable convolution, we also report the performance of Panoptic-FPN-D, i.e., the deformable counterpart of Panoptic-FPN. As can be seen, applying deformable convolution in the semantic segmentation branch gives limited improvements. We keep the number of flops and parameters of Auto-Panoptic backbone the same as that of ResNet50 [17] and DetNAS [32] for a fair comparison. Compared to previous handcrafted pipelines, our Auto-Panoptic achieves new state-of-the-art results in terms of PQ on two large-scale benchmarks. On COCO, Auto-Panoptic surpasses SOGNet by a large margin, i.e., 1.1 in terms of PQ. On ADE20K, Auto-Panoptic surpasses BGRNet by 0.6 in terms of PQ. This demonstrates that the proposed framework is capable of finding a more efficient model for panoptic segmentation compared to hand-crafted ones. The searched architecture and more visualizations can be found in Supplementary Materials.

## 4.4 Ablation Studies

**Effectiveness of Auto-Panoptic Search Algorithm.** To eliminate the effect of the search space, we compare Auto-Panoptic to different models generated from the adopted search space in Table 3. (1)

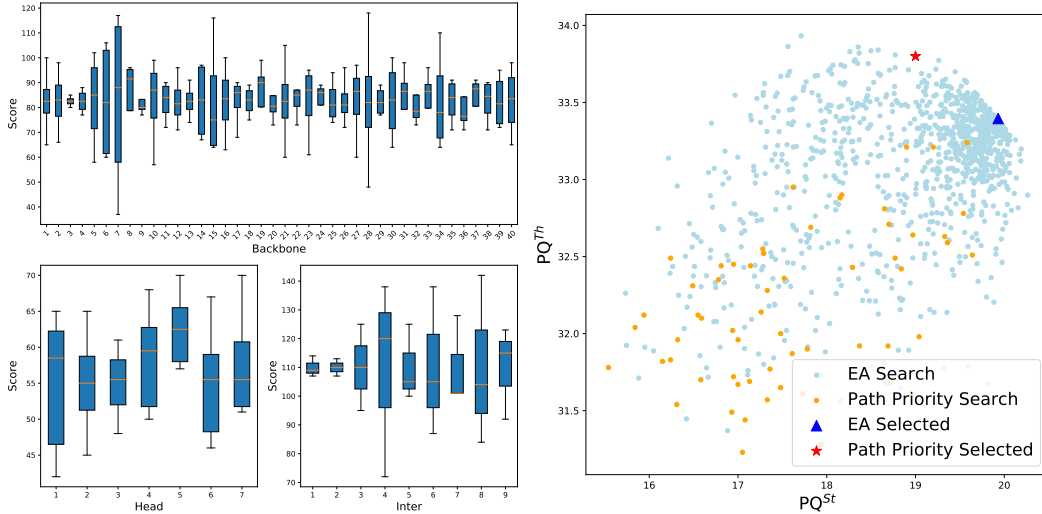

(a) Scores of each layer after Path-Priority Policy.　　(b) Models explored by Path-Priority Policy and EA.

Figure 2: Statistics of Path-Priority Search Policy. (a) The scores of different operators in most layers are dispersed and discriminative. (b) Our Path-Priority Search Policy can discover a more optimal model (44.8 vs 43.8 PQ) by exploring much fewer models compared to EA (60 vs 1000).

We pick 5 random models and retrain them with the same setting as Auto-Panoptic. The best of them achieves 41.1 PQ, which is 3.7 PQ lower than Auto-Panoptic. This is not surprising since the combinations of different component architectures can have a significant impact on the performance. (2) We use the searched backbone in DetNAS [32] along with the searched head in Auto-Panoptic to build the hand-crafted model in Table 3, which is 1.7 PQ lower than Auto-Panoptic. (3) We replace all convolution layers in the searched head with 3x3 convolution, referred as Auto-Panoptic-Plain, and get 42.9 PQ, which is 1.9 lower than Auto-Panoptic. These results demonstrate the effectiveness of the proposed search space as well as the effectiveness of Auto-Panoptic after eliminating the effect of search space.

Table 3: Effectiveness of Auto-Panoptic search algorithm.

| Model | PQ |
|---|---|
| Random | 41.1 |
| Hand-crafted | 43.1 |
| Auto-Panoptic-Plain | 42.9 |
| Auto-Panoptic | 44.8 |

Table 4: Effectiveness of Cooperative Multi-Component Architecture Search.

| Backbone | Head | Inter | PQ | $PQ^{Th}$ | $PQ^{St}$ |
|---|---|---|---|---|---|
| ✓ | | | 43.1 | 50.5 | 32.0 |
| | ✓ | ✓ | 43.3 | 50.0 | 33.2 |
| ✓ | ✓ | | 43.9 | 50.2 | 34.5 |
| ✓ | ✓ | ✓ | 44.8 | 51.4 | 35.0 |

**Effectiveness of Multi-Component Architecture Search.** We explore the impact of separate components in the panoptic pipeline to verify the effectiveness of Multi-Component Architecture Search. We replace some specific parts of the baseline model with corresponding Auto-Panoptic components and report the performance on COCO val set. As can be seen in Table 4, the models with one or two components replaced by searched backbone or head or inter-branch fusion module are inferior to the full version of Auto-Panoptic. This demonstrates the point that the performance of panoptic pipeline can be greatly improved when the key components are in an optimal configuration. Moreover, the performance increases when more parts are searched, which manifests the superiority of our Cooperative Multi-Component Architecture Search.

**Effectiveness of Path-Priority Search Policy.** We compare Path-Priority Search Policy with conventional methods, i.e., random search and EA in Table 5. For random search, we randomly enumerate 300 models from the whole search space and report the best model found. It exhibits poor performance (42.8 vs 44.8 PQ) since it is hard to explore the whole search space without any heuristics. For EA, we follow [32] and maintain a population of 50 individuals, and repeat the evolution process

for 20 iterations. As can be seen from Table 5, EA is capable of picking out competitive models but fails to match our Path-Priority Policy from the aspect of both performance (43.8 vs 44.8 PQ) and cost (1000 vs 60 models). We further visualize the score distribution of each layer after executing Path-Priority Search Policy in Figure 2a. As can be seen, the scores of different operators in search space have clear discrepancies in most layers. The models explored by EA and Path-Priority Search Policy can be found in Figure 2b. We find that the numerous models explored by EA evolve over time and are relatively concentrated, while Path-Priority Search Policy is able to find a more optimal architecture with much fewer dispersed models.

Table 5: Comparisons between Path-Priority Policy and conventional methods of architecture search.

| Methods | #Evaluated | PQ |
|---|---|---|
| Random Search | 300 | 42.8 |
| EA [15] | 1000 | 43.8 |
| Path-Priority | 60 | 44.8 |

Table 6: Transferability of searched architectures.

| Source | Target | PQ | $PQ^{Th}$ | $PQ^{St}$ |
|---|---|---|---|---|
| ADE20K | COCO | 43.9 | 50.1 | 34.6 |
| COCO | COCO | 44.8 | 51.4 | 35.0 |
| COCO | ADE20K | 32.0 | 33.4 | 29.1 |
| ADE20K | ADE20K | 32.4 | 33.5 | 30.2 |

**Transferability of the Searched Architectures.** To verify the transferability of Auto-Panoptic, we perform architecture search on the source dataset and retrain the best model found on the target dataset. The results of transfer experiments are shown in Table 6. We find that Auto-Panoptic can transfer well among different datasets. We attribute the robustness of Auto-Panoptic to the superiority of the strictly fair training strategy and Path-Priority Search Policy since the choice paths at each layer are equally treated throughout the training and architecture search process.

## 5  Conclusions

In this work, we extend the common single-task NAS into the multi-component scenario and propose Auto-Panoptic which makes the first attempt to search for all key components of the panoptic pipeline with an efficient intra-modular search space and problem-oriented inter-modular search space. Furthermore, we significantly reduce the computation in the architecture search phase using the proposed Path-Priority Search Policy and obtain better performance compared to EA. Our Auto-Panoptic achieves new state-of-the-art results on COCO and ADE20K.

## Broader Impacts

This work makes the first attempt to search for all key components of panoptic pipeline and manages to accomplish this via the proposed Cooperative Multi-Component Architecture Search and efficient Path-Priority Search Policy. Most related work in the literature of NAS for fine-grained vision tasks concentrates on searching a specific part of the network and the balance of the overall network is largely ignored. Nevertheless, this type of technology is essential to improve the upper bound of popular detectors and segmentation networks. This may inspire new work towards the efficient search of the overall architecture for fine-grained vision tasks, e.g., object detection, semantic segmentation, panoptic segmentation and so on. We are not aware of any imminent risks of placing anyone at a disadvantage. In the future, more constraints and optimization algorithms can be applied to strike the optimal trade-off between accuracy and latency to deliver customized architecture for different platforms and devices.

## Acknowledgments and Disclosure of Funding

This work was supported in part by National Key RD Program of China under Grant No. 2018AAA0100300, National Natural Science Foundation of China (NSFC) under Grant No.U19A2073 and No.61976233, Guangdong Province Basic and Applied Basic Research (Regional Joint Fund-Key) Grant No.2019B1515120039, Nature Science Foundation of Shenzhen Under Grant No. 2019191361, Zhijiang Lab's Open Fund (No. 2020AA3AB14).

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
