[Supplementary Material]

# Supplementary Materials of Auto-Panoptic: Cooperative Multi-Component Architecture Search for Panoptic Segmentation

**Yangxin Wu**[1], **Gengwei Zhang**[1], **Hang Xu**[2], **Xiaodan Liang**[1,3], and **Liang Lin**[1,3*]

[1]*Sun Yat-sen University*, [2]*Huawei Noah's Ark Lab*, [3]*DarkMatter AI Research*
wuyx29@mail2.sysu.edu.cn, {zgwdavid, chromexbjxh, xdliang328}@gmail.com, linliang@ieee.org

## 1   Details of Search Space

We use the same backbone search space as DetNAS [3] and the details are listed in Table 1. The first block of each stage has stride 2 for downsampling. There are 40 blocks in total except for the stem. There are 4 choice paths in each ShuffleNetv2 block: the block with kernel size $3\times3$, $5\times5$, $7\times7$ or the block replacing the right branch with three separable depthwise $3\times3$ convolutions, i.e., an Xception block. The total number of parameters in this backbone is 1.3G, and we follow DetNAS [3] to rescale the channel configuration to a larger one (3.8G) after the architecture search phase to meet the parameters amount of commonly used ResNet50 (3.8G) for a fair comparison.

Table 1: Details of backbone search space.

| Stage | Block | Channel | #Conv |
|---|---|---|---|
| 0 | Conv3×3-BN-ReLU | 48 | 1 |
| 1 | ShuffleNetv2 block (search) | 96 | 8 |
| 2 | ShuffleNetv2 block (search) | 240 | 8 |
| 3 | ShuffleNetv2 block (search) | 480 | 16 |
| 4 | ShuffleNetv2 block (search) | 960 | 8 |

## 2   Details of Experiment Setup

The loss weights for foreground branch (including classification head, box head, and mask head) and semantic segmentation branch are 1 and 0.3, respectively. Following [3, 1], we recalibrate BN statistics for each single-path model before architecture search phase since the batch statistics of each model should be independent of others. We use a small subset of images from the training set, i.e., 500 images, and forward them in the network to get appropriate running mean and variance values. Given the instance segmentation outputs and semantic segmentation outputs, we use a simple heuristic method similar to Panoptic-FPN [2] to derive the panoptic outputs. We first filter out the instance segments that meet one of the following conditions: 1) segments with confidence scores lower than 0.5; 2) segments that are covered by other segments with higher confidence and the overlap ratio exceeds 0.5; 3) segments with the area less than $64 \times 64$. After filtering, we sort the segments with the order of increasing area and map them onto the canvas one by one. When all instance segments have been processed, we resolve the overlap between instance segments and background segments in favor of instances.

Figure 1: The searched architecture on COCO.

# 3 Searched Architecture

We visualize the searched architecture on COCO in Figure 1. For brevity, some unsearched parts (e.g., FPN, final conv in semantic segmentation branch, etc) are not shown.

# 4 Visualizations

We visualize the results on COCO and ADE20K in Figure 2 and Figure 3.

Figure 2: Visualizations on COCO.