[Reviews · NeurIPS 2020]

Review 1

Summary and Contributions: The paper presents a NAS-based attempt to search for panoptic segmentation architecture. The search space is the backbone layers (same DetNAS), 4 layers in instance mask head, 3 layers in semantic head, and 9 integer parameters in the thing/stuff connections. More efficient search is proposed that beats evolution algorithm. Experimental evaluation is done on COCO and ADE20k with solid improvements.

Strengths: * The search algorithm seams very efficient for the setup of layer selection * The final performance is great * Ablation experiments answers most of the questions * Transferability experiments are useful

Weaknesses: * 12 epoch training for COCO is known to result in severely underfitted models. It may be the case that the search is finding the model that is slightly faster to train than others but with 36 epochs (3x schedule) the model with DetNASNet backbone and with 5x5 DFConvs in both heads will match or even surpass the performance of the best found model. * I found the search for the ratio parameters in the Inter-Modular Attention module to be likely an overkill. How much worse is it to have 2 ratios (one for b->f and one to f->b) and treat them as a hyper-parameters? How much worse the best model with such hyper-parameters will be than the found one? I'm not sure the search space presented in the paper is really needed and, hence, the claims about new "problem-oriented inter-modular search space" seem like an oversell to me. * I'm curious why the author did not include Cityscapes dataset in their experimental evaluation? Most of the methods that the paper is comparing against use it along with COCO and the code is readily available for many of this models. Is it too small for the NAS to work properly? I think if it is, such negative result should be included in the paper.

Correctness: L152 states: "In contrast, Auto-Panoptic aims to capture the reciprocal relations between things and stuff at the feature level rather than the mask level, which allows us to circumvent the complicated process of instances with variable number and size." However, Figure 3 in the supplement clearly shows that combination of things and stuff predictions are done sub-optimally and there are holes between masks. While the attention mechanism is helping I don't think it supports the claim that post-processing is not needed anymore and the model presented does not require any alignment between things and stuff.

Clarity: The paper is written clearly and was easy to follow up to the experimental evaluation. Some details of exact setups are not fully clear: - were Deformable convs added to both mask and segmentation branch of Panoptic FPN? - descriptions for hand-crafted and Auto-Panoptic-Plain rows in Table 3 are very short and I am not sure I understand them correctly. I think this experiment is the most important in the paper and should be clarified. I encourage the authors to add the clarification to their rebuttal. - inter-modular attention connects corresponding layers of RPN and Segmentation branch (for example, stride 4 with stride 4) according to (1) and (2). L161 states that there are 4 RPN level and 4 levels from semantic branch. Based on that, I assumed that the top level of RPN with stride 64 is ignored and only 8 ratios are searched. However, in L232 L_2=9 and figure 1 in the supplement has 9 ratios. I was not able to explain this discrepancy. - is BatchNorm frozen during re-training?

Relation to Prior Work: I'm not an expert in NAS literature and I'll leave this judgment to my fellow reviewers. Panoptic segmentation literature is survived properly. I would recommend the authors to relate their inter-modular attention module to the proposal relation module from AUNet [23]. Although not exactly the same, the two mechanism are quite similar a relation in the text will be beneficial.

Reproducibility: Yes

Additional Feedback: * How much slower is the new model in comparison with baseline model with the DetNASNet backbone? ********************************************************** I read the other reviews and the rebuttal carefully and I appreciate the authors answers to some of my questions. I still have a concern regarding the proposed intra-modular component. The search space of the reduction factors per channels seem enlarged without a need for it. I doubt a single factor would work much (if at all) worse. Unfortunately, in the rebuttal the authors did not provide the evidence that different reduction factors for different levels do make a difference. Instead, another experiments with shared parameters was provided. Overall, I think the NAS-based exploration of specific panoptic segmentation-oriented models is a very interesting direction. Our community has suggested diverse ways to connect the stuff and things branches (BANet or BGRNet) or different unified models (Panoptic-deeplab or DETR) for panoptic segmentation. There is a chance that this paper will pave a way to the exploration, therefore I keep my rating but I'm really not that confident.


Review 2

Summary and Contributions: The submission presents a system for neural network architecture search for the components specific to the Panoptic Segmentation problem. Most Panoptic Segmentation models will at some point have two "branches" that each specialize in either the the "thing" classes (car, person, etc.) that have discrete instances and the "stuff" classes (grass, sky, etc.) that do not. What this paper deals with is searching the architectures of each branch (intra-modular search) as well as operations that can share information between the branches (inter-modular search). This is done with a "one-shot" architecture search that learns and then samples from a "supernet" that contains a superset of the connections in any candidate architecture. Search and evaluation both are done on COCO and ADE20k datasets.

Strengths: i) On the COCO val set, it beats methods that placed highly (on the test set) in the most recent COCO challenge. This is at least competitive with the state of the art, though it is hard to be certain as the submission only compares to other SOTA methods on the validation set (in Table 2), when test-dev PQ may be the more authoritative comparison. ii) The idea of sharing more levels of features between "thing" and "stuff" branches is particularly interesting, and doing architecture search on the ways to do the sharing is a great way to explore this. It is a uniquely well-suited and important application of neural architecture search methods. iii) Empirical evaluation is fairly well-executed, with some large-scale experiments and strong ablation and partial comparison results. iv) Code is provided and looks to be quite complete. The full training and testing code is included. The search method itself included (see under 'code/maskrcnn-benchmark/maskrcnn_benchmark/engine/architecture_search.py').

Weaknesses: v) It is difficult to attribute the source of empirical gains, since the paper is presenting both a problem-specific architecture search space and a particular search method. The comparison to random is missing some potentially-important measures as it has no error bars or plot of the distribution. Though the comparison to evolutionary methods in Fig 2. is a good experiment along these lines, the (missing) random comparison is especially important [a]. The comparison to random is against the *best* model found by random search, instead of error bars or any modeling of the search space. This'd be important for comparisons that separate out the search vs design space as in [a,b]. vi) There also does not appear to be multiple end-to-end trials with different random seeds (including e.g. a full supernet training) of the proposed method. vii) The submission is not a strong contribution to the wider general NAS literature. The central novelty is in the panoptic-segmentation-specific search space. The search method is closely based on [8,15]. Comparisons to other practical NAS methods are limited to partially replacing one part of the method with an evolutionary search. I'd guess the submission will be of a high degree of interest only to segmentation and vision communities (admittedly very large communities). I don't see an insight or innovation that's likely to be adopted for other ML problems. viii) Unfortunately, the code is not that well organized (e.g. main code under nested maskrcnn-benchmark/maskrcnn_benchmark folders). There are not a lot of technical documents or comments, and running it has many dependencies on a particular AWS environment that may be difficult to untangle. I don't want to penalize the authors for including code simply because the code is undocumented and has unclear organization, as nearly all research code suffers from this, so this is not an important point for the score. ix) Comparisons are done between evolutionary search, random search, and the provided method. Is the code for the evolutionary and random search also included in the submission? I did not find it, but it is reasonable to presume these comparisons were done in a consistent way, as the authors performed these searches themselves. [a] Li & Talwalkar. "Random search and reproducibility for neural architecture search." UAI 2019. [b] Radosavovic et. al. "On Network Design Spaces for Visual Recognition." ICCV 2019.

Correctness: The following is meant to highlight relevant points in the checklist for NAS reproducibility from Lindauer & Hutter (https://www.automl.org/wp-content/uploads/NAS/NAS_checklist.pdf). x) Complete code with hyperparameters is provided. I don't see the random seeds used in experiments specified anywhere obvious. The only explicit seeding I found is in some of the Mask R-CNN unit tests. xi) The evaluation protocol appears to be consistent up to the code used to evaluate the segmentations: the official evaluation code by Kirillov is included the supplementary under 'code/maskrcnn-benchmark/panopticapi'. xii) Relevant missing comparisons are the "performance over time," the full random comparison, and multiple runs as described in #5 previously. xiii) The process for deciding search hyperparameters is not described completely, for instance stating "T = 5, E = 12 suffices to find a satisfactory model." xiv) Wall-clock time to execute the NAS method is not reported, but most of the related details are also provided, such as the number of epochs the supernet was trained and the number of models evaluated by the path-priority search. Overall I'd judge this is a slighly higher score on the checklist than I have seen from most accepted architecture search papers.

Clarity: The submission is reasonably readable. Key concepts are introduced and built up in a good sequence.

Relation to Prior Work: There is not a lot of closely comparable work. The submission deals with some issues specific to architecture search on panoptic segmentation problems, and I am not aware of other publications in this space. There is separately reasonable coverage of the panoptic segmentation literature and NAS, the latter including citations to NAS work on other segmentation problems. The authors do a good job of discussing this prior work and why their problem differs.

Reproducibility: Yes

Additional Feedback: Grammar and wording suggestions: 12: "the advantages" -> "advantage" 37: "on graph" -> "on a graph" 40: "backbones and heads" 41: add "a" before "panoptic" 57: Should there be an article before "supernet?" 63: network -> networks 76: add "a" before "semantic" 78: remove "with" 100: add "a" before "panoptic" 102: pipeline -> pipelines 110: add "the" before "backbone" 131: "the" before "box head" 132: "the" or "a" before "mask head" 164: Either "are channel-wise statistics" or "a channel-wise statistic" 169: controls -> control 233: is -> are 238: "the" before "training" and "validation" 257: "the" before "search" 295: "a" or "the" before "panoptic" 295: "an" before "efficient" After rebuttal: I mostly disagree with the point in the rebuttal that "Hand-crafted model in Table 4 [meant to be 3?] clearly demonstrates the effectiveness of the search algorithm." The hand-crafted algorithm doesn't separate the effect of space vs search as rigorously as the kinds of experiments described in [a,b]. However, the described improved comparison to random might do it. The rebuttal doesn't strengthen the case that there is a contribution to wider NAS literature: problem-specific search spaces without a new algorithmic component, or an insight that may apply to other search spaces, will be of limited value to other problems. So I do not raise my score, but continue to weakly recommend acceptance: the paper is reasonably well-executed and the contribution to the segmentation/vision literature makes it valuable.


Review 3

Summary and Contributions: The paper presents a new panoptic segmentation model that is designed by using a neural architecture search (NAS). It searches for three key components (backbone, head, inter-branch) of the panoptic pipeline. The author reduce the computation in the architecture space using a path-priority search policy. The final model achieved favorable results on two standard benchmarks: COCO and ADE20K

Strengths: + This paper is the first attempt to use NAS on the panoptic segmentation task. + The author well formulate the search space of the panoptic segmentation model. + The proposed path-priority search reduces the time complexity of conventional architecture search compared to the evolutionary algorithm (EA).

Weaknesses: - The important architectural designs for panoptic segmentation lie in 1) how we build interaction between instance or semantic segmentation head, 2) how we deal with the occlusions (e.g., instance-instance, instance-semantic), and 3) how can we make the instance and semantic predictions consistent. However, the current method mainly focuses on the backbone design, which is out of the track of the community's interest. - The proposed inter-modular search attempts to explore the interaction between instance or semantic segmentation head to some extent. However, the search space is too simple compared to recent advanced interaction strategies (refer to the BANet or BGRNet). Why not adopting these strategies in the search space? - The algorithm box is hard to follow, reducing the reproducibility of the proposed method.

Correctness: The arguments and experiments are solid.

Clarity: The paper is easy to follow.

Relation to Prior Work: The related work section well describes the distinct points from previous works.

Reproducibility: No

Additional Feedback: The paper has value on applying the NAS firstly to the panoptic segmentaiton task. However, I think the current architecture search should focus more on the head design that can solve distinctive panoptic segmentation issues not the backbone design. --After Rebuttal I checked out the authors' rebuttal and other reviewers' comments. I have the following remaining concerns. 1. Technical novelty is limited (Search Algorithm) I see the main search algorithm is identical to [15]. More specifically, both the supernet-then-architecture search scheme and the uniform sampling strategy [8] are already introduced and verified in [15]. The proposed path-priority policy is new, but its contribution is relatively weak to the previous NAS algorithms (also noted by R2). Moreover, it is not clear whether it is indeed general (also noted by R2 and R4). (Search Space) The authors mainly focus on module/cell-level search (i.e., different convolution layers/reduction ratios). I think this setup prevents to reflect more interesting task-specific attributes in the final model. To address this issue, I believe network-level search space, which includes panoptic segmentation-specific domain knowledge, should be designed and investigated in parallel. Here, the authors simply combine well-known (panoptic) network designs from previous literature. For example, the backbone is mainly based on ShuffleNetv2 [30] for efficiency. The things and stuff heads are from PanopticFPN [20]. The formulation of inter-modular attention is similar to AUNet [23]. 2. Clarity The proposed path-priority (Algorithm 1) is not clearly elaborated, and thus is hard to follow and understand why it works effectively (also noted by R4). On the positive side, the paper firstly attempts to apply NAS on panoptic segmentation and the empirical results are promising. All the reviewers agree that this may obviously invigorate follow-up researches. In that regard, I would like to raise my score from 5 to 6.


Review 4

Summary and Contributions: This paper proposes Auto-Panoptic, which is a Neural Architecture Search(NAS) work performed directly on panoptic segmentation task. By searching on backbone, mask/semantic segmentation head and an additional inter-modular attention, with the proposed effective searching policy, the model can achieve state-of-the-art performance on panoptic segmentation task for both COCO and ADE20K dataset.

Strengths: The performance of this paper is very good, the experiment details are provided thoroughly. The proposed inter-modular attention is also interesting. NAS problem is general enough and related to NeurIPS community, architecture searching directly on target task like panoptic segmentation is also worth investigating.

Weaknesses: Please see correctness, clarity, and additional feedback for detailed comments.

Correctness: The technical novelty of this paper mainly comes from the Path-Priority Search Policy beside inter-modular attention. However, the description of this part is very limited, there is no analysis/explanation about why this will work well. Although the authors provide some intuition like "it is possible to rate different choice paths rather than an entire model in order to reduce the complexity of architecture search", from my understanding the evaluate/rank process in Architecture Search Phase in Algorithm 1 will not affect the permute (sampling) process. I do not think it matches the text in the paper.

Clarity: The RANK function in Algorithm 1 is not clearly stated. the "if |P_1|| I and |P'_1| i and |P_2|| i then" clause is very confusing and I don't get the meaning. Algorithm clarity issue is showed in the correctness part above.

Relation to Prior Work: This paper extends DetNAS with a different search policy and apply the proposed method to panoptic segmentation, the similarity and difference are discussed clearly.

Reproducibility: Yes

Additional Feedback: I think Hand-crafted baseline in Sec 4.4 makes less sense. Instead, a more convincible is that fixing the backbone as the DetNAS one during supernet training/architecture search phase, to show that searching all component is necessary indeed. For me it would be natural to include FPN and RPN/Detection head as the searched components to further boost performance, why the authors decide not to do this confused me. The x and y axis value in Fig. 2b seems abnormal, they are significantly lower than numbers reported in the tables. My rating is 5 mainly because of the issue I mentioned in correctness. Although this paper achieves good performance, the writing makes it hard to understand the mechanism behind the approach and I don't think the current version is ready for publication. I'm happy to raise my rating if the authors can elaborate it clearly in the rebuttal. ---- Post rebuttal: I've read other reviewers' comments and the authors' response. Although this paper achieves good results, the technical novelty part is somewhat limited. The authors elaborate how the path-priority policy works in the response. However I still do not get why such greedy search policy works well and it seems to simple to generalize well from my perspective. I also do not think the search space is carefully considered to fit the panoptic segmentation task (also noted by R1/R3). To me the authors fail to show that why their method is unique/important to panoptic segmentation, or they just choose a less competitive task that NAS is hardly studied. Given the fact that this search policy works well on panoptic segmentation indeed. I raise my rating to 6

[Author Response · NeurIPS 2020]

We thank reviewers for the constructive comments. We are encouraged that our paper presents very efficient search
algorithm [**R1**,**R2**,**R4**], our performance is significant [**R1**,**R2**,**R4**], the proposed search space is well-formulated
and uniquely well-suited [**R2**,**R3**], our ablations are well-executed [**R1**,**R2**,**R3**], NAS for multi-component scenarios
is worth investigating [**R4**], and our codes are complete with good reproducibility [**R1**,**R2**,**R4**]. All comments are
addressed and technical details are provided below. We will release all models and further polish the documents.

**Inter-modular search space** [**R1**] We design a fine-grained inter-modular search space mainly because (i) there is
a large discrepancy on the distribution between features of different levels from both branches and a level-to-level
attention mechanism enables **input-specific** and **semantic-aware** feature calibration across branches, (ii) the proposed
operation is highly efficient and incurs minor increase in parameters. To verify this, we keep only **vector** $a_{b\to f}$ and
$a_{f\to b}$ (shared among levels) via Eq.1 after upsample the multi-level features in both branches, and repeat the overall
searching process. The result on COCO val set is 44.0 PQ (+0.1 PQ vs no inter-modular, -0.7 PQ vs Auto-Panoptic).

**Cityscapes**[**R1**] Due to time limit, we reuse DetNAS backbone (Params/Flops same as **R50**) and execute **partial**
**search** and achieves 59.8 PQ, which is satisfactory and is expected to obtain more performance gain in the full version,
i.e., overall search with proper training protocols/hyper-parameters and larger backbone. We do not observe degradation
under UPSNet setting (12K iter, imagenet pretrain) and **do not** use tricks like extra long training (EfficientPS, Panoptic-
FPN, Seamless), more powerful pretrain (EfficientPS), delicate post-process (AUNet,BANet) and ms testing.

**Longer training** [**R1**] Under 3x schedule, our Auto-Panoptic achieves 45.2 PQ, while DetNAS backbone with 5x5
DF conv in both heads achieves 44.8 PQ, which is 0.4 PQ lower and is much slower due to the heavy head. This
demonstrates the superior performance of Auto-Panoptic is not because 'it is slightly faster to train'.

**Contribution to NAS literature**[**R2**,**R3**] We want to emphasize our work aims to overcome immense amount of
computation and severe convergence inefficiency when extending the current single-task NAS to the more complicated
multi-component scenarios (extra large search space 6.7e33 vs DetNAS's 1.2e24). To achieve this, we propose
customized search space (including problem-oriented inter-modular search space that enables flexible alignment across
branches) and a highly efficient Path-Priority Policy ($\times 16.7$ speedup with better performance).

**Comparison to random baseline**[**R2**] The error bars of the random baseline for 5 trials is (40.46$\pm$0.67). We do not
have multiple trials due to the cost and will add it in the revised version. For eliminating the effect of search space, the
Hand-crafted model in Table 4 clearly demonstrates the effectiveness of the search algorithm.

**Hyperparamter**[**R2**, **R3**] $E = 12$ is the least common multiple of $|P_1|, |P_2|, |P_3|$. Enumerating 12 models per cycle
can ensure each path in different component has the same number of evaluation. $T = 5$ makes the enumerated models
in Path-Priority roughly equals to that of one generation in EA (60 vs 50).

**Alg.1/Fig.2 Clarification**[**R3**,**R4**] Statement '$|P_1|$ | $i$ **and** $\cdots$': $i$ is divisible by $|P_i||_{i=1:3}$. This ensures each path in
each component is trained equally. RANK function: we rank and score the paths according to their fitness, i.e., PQ. The
scatter in Fig.2b plots the performance of enumerated models before retraining, while the best model found is retrained
(c.f. Sec.4.1), thus these enumerated models have lower PQ since they are not fully trained.

**Path-Priority (abbr. PP)**[**R4**] The evaluate/rank process will not affect the sampling process. Consider a layer with
6 paths, EA uses uniform sampling and the probability that a path is not sampled after $N$ samplings is $(\frac{5}{6})^N$. When
$N = 6$, PP based on **fair sampling without replacement** can ensure all paths are evaluated exactly once, while in EA
the probability of a path for not being sampled is 33.5%. Thus PP can cover the whole search space with much reduced
computation. We use the PQ of the model as the performance indicator of the paths based on a greedy assumption: when
the model is good, then its paths are good as well. Thus we assign the score of the paths based on $score_i = K - rank_i$
and **accumulate the scores over** $T$ **cycles**. The paths with the highest score are picked to build up the best model.

**Plain/Hand-crafted baseline**[**R1**,**R4**] For Auto-Panoptic-Plain, we replace all layers in the searched head with 3x3
conv, which is to verify the effectiveness of the proposed search space for head. Regarding the Hand-crafted baseline,
we follow R4's suggestion to fix DetNAS backbone and search for head architecture, which achieves 43.4 PQ on COCO
val set. This result together with Table 3 demonstrate the effectiveness of Multi-Component Search.

**Searching FPN/RPN/Detection Head** [**R4**] For FPN, all layers are 1x1 conv and are used to adjust the channel number,
they act more like fc layer slide over the feature map. There is only 1 conv in RPN and no conv in detection head (2fc
box head). Adding them to the search space may bring limited gain and we do not search them for simplicity.

**Inference time/Wall clock time**[**R1**,**R2**] Inference: Auto-Panoptic(186ms) vs DetNAS-backbone-with-searched-head
(209ms) vs UPSNet (171ms). Wall clock: supernet fine-tuning: 2 days, Path-Priority: 0.4 days (on 8 GPUs).

**Experiment setup**[**R1**] DF conv is not added to Panoptic-FPN (c.f. Table 2). We build inter-modular search space
between the 5-th RPN level between the last level of semantic branch. We do not freeze BN during retraining.

[Meta-Review · NeurIPS 2020]

After rebuttal, there is a consensus among reviewers for weak acceptance. Although their remains room for improvements, the AC agrees that the NAS-based approach for panoptic segmentation architecture search is interesting, and therefore recommends acceptance. The authors are highly encouraged to update the final version of the paper based on reviewers' comments.